# The Mediating Effect of Subjective Housing Quality on the Relationship Between Housing Conditions and Mental Health: Evidence from China’s Mega-Cities

**DOI:** 10.3390/bs15040485

**Published:** 2025-04-07

**Authors:** Hao Yuan

**Affiliations:** School of Sociology and Political Science, Shanghai University, Shanghai 200444, China; yuanhao@shu.edu.cn; Tel.: +86-21-66134142

**Keywords:** housing area, home ownership, housing environment, mental health, subjective housing quality, Chinese mega-cities

## Abstract

This study examines the mediating effect of subjective housing quality between housing conditions and mental health, using survey data from ten Chinese mega-cities. The results from multi-level linear regression models show that housing areas are highly associated with subjective housing quality and that renters have lower levels of subjective housing quality than homeowners. At the community level, the age of housing tends to diminish its subjective quality, while a lower plot ratio is associated with more favorable evaluations of housing conditions. Surprisingly, educational resources in proximity to housing are negatively associated with subjective housing quality. Subjective housing quality is closely linked to mental health. Additionally, the results show that home ownership significantly strengthens the association between subjective housing quality and mental health. Namely, the mediating effect of subjective housing quality on the relationship between housing conditions and mental health is stronger for homeowners than for renters.

## 1. Introduction

With the rapid development of urbanization in China, housing problems in mega-cities have become increasingly prominent and have comprehensively affected residents’ quality of life. Domestic migrants and young people are burdened with significant financial pressures when it comes to purchasing or renting housing. Meanwhile, a large number of old housing units built before the year 2000 are small in size, poor in quality, and incomplete in terms of ancillary facilities; thus, the demand for improved housing is becoming increasingly acute in China ([17]).

Existing studies have confirmed the link between housing conditions and mental health ([1]; [25]). Physical housing features and home ownership are the major factors directly impacting mental health problems ([25]).

These housing disadvantages are not only an established objective fact, but are also closely related to subjective consciousness. How do residents in China’s mega-cities currently perceive the quality of their housing? And how do the evaluations of their own housing consequently mediate the association between objective housing conditions and mental health? In order to answer these questions, this study uses a survey sample of 5032 residents in Chinese mega-cities, namely Beijing, Tianjin, Shanghai, Hangzhou, Guangzhou, Shenzhen, Chongqing, Chengdu, Wuhan, and Changsha, and examines the associations of housing conditions and community environments with subjective housing quality and mental health.

## 2. Literature Review

Housing holds a pivotal and crucial position in Chinese family life ([6]). Individuals often dedicate a substantial portion of their lives to their homes and readily incur financial expenses to ensure their families’ well-being. Housing shortages, housing burdens, and a poor community environment can all exert a profoundly negative impact on people’s mental health ([10]). The mediating effect of subjective housing quality in the relationship between housing conditions and mental health is not fully understood. Individuals may hold diverse aspirations for their dream homes, resulting in varied subjective evaluations of their current housing quality. Residents of specific communities may form distinct perceptions concerning the quality of their living spaces, which can significantly influence their mental health ([4]).

Over the past three decades, China’s urban housing reforms have shifted from a socialist administrative distribution system to a market-driven system of housing development and consumption ([27]). This transition has been accompanied by a significant rise in housing prices, which has led to increased housing shortages and social segregation. Chinese culture has always emphasized the importance of “living and working in peace and contentment”; thus, housing can reflect the status of an individual or even the entire family ([5]; [8]; [14]). Britons generally believe they should own a home by the time they turn 30 ([11]). Based on a study of middle-class Australians, [7] ([7]) argue that home ownership is the fulfillment of the “Great Australian Dream” and that owning a home imposes a positive effect on individuals’ social status, as well as their sense of identity and belonging. In China, owning a home provides people with important access to credit facilities and helps them to enhance their well-being ([29]).

Many existing analyses focus on subjective housing quality ([23]; [20]). Subjective housing quality reflects an individual’s perception of the quality of their housing ([20]). In previous studies on the home-owning class, scholars have usually adapted objective indicators such as housing area, home ownership, the number of houses, and the payment required for house purchase ([27]). The possession of a home is an important factor affecting residents’ evaluations of their housing status ([4]). As more real estate or larger houses mean a higher level of wealth accumulation and appreciation, they are consequently highly associated with individuals’ quality of life. Better housing conditions are associated with higher levels of subjective housing quality ([20]). Correspondingly, this study predicts that objective housing conditions, such as home ownership, housing area, and home environment, may have significant impacts on subjective housing quality.

The quality of home environments, encompassing aspects such as accessibility, affordability, and the availability of essential facilities like plumbing and cooking, plays a crucial role in enhancing subjective housing satisfaction and mental health ([10]; [20]; [24]). In mega-cities, collective consumption reflects the complex relationships between private housing and public resources, ultimately resulting in the stratification of subjective housing quality ([34]).

During the ongoing process of urbanization in China, certain communities located far from city centers, as well as older communities in downtown areas, frequently experience economic disadvantages. These communities often suffer from inadequate amenities, a lack of leisure and entertainment facilities, inconvenient transportation, poor housing conditions, and unfavorable natural surroundings ([13]). Furthermore, they have limited access to shopping centers and medical services. These disadvantages might reduce residents’ perceptions of their subjective housing quality and further undermine their mental health in a direct or indirect way. Community-level variables are the most influential predictors for understanding variations in residential satisfaction ([31]).

Various types of communities in mega-cities are spatially intertwined. Many communities in city centers or downtown areas, although they are conveniently located and well-equipped with surrounding amenities, tend to have higher housing prices, dilapidated housing and living facilities, and crowded community environments. Meanwhile, the residents in these disadvantaged communities may have lower expectations and demands for housing quality and pay less attention to their residential environments. In these communities, many residents view their homes merely as places for sleeping and resting, and they are less concerned about the quality of their houses.

Moreover, scholars also emphasize that home ownership may amplify or attenuate the impacts of housing on mental health. In Chinese society, the establishment of a family is closely related to housing, and “owning one’s own home” is the obsession of a vast majority of Chinese people ([4]). Renters may be particularly susceptible to the ill effects of low-quality housing ([10]). This group may be especially vulnerable because of social exclusion caused, in part, by their inability to participate in community governance. It is widely believed in China that renting a flat provides no sense of family and belonging, and instead burdens renters with many problems, such as high rent, poor stability, low comfort, etc. Only owning a house ensures that residents can truly settle down and enjoy a relaxing home. In most Chinese mega-cities, homeowners have some privileges over renters. Namely, renters do not have the same rights in terms of access to public services ([16]). This discrimination may further diminish renters’ subjective perceptions of housing quality. Homeowners usually invest a substantial amount of money in their homes, often incurring significant mortgages ([17]). Understanding the disparities in subjective evaluations of housing conditions between renters and homeowners is essential for grasping the association between housing quality and mental health.

## 3. Methods

### 3.1. Study Design

To achieve the research objectives, the research process primarily involved the development of a cross-level evaluation framework on housing environment, subjective housing quality, and mental health. The Social Survey Center at Shanghai University conducted the “Survey on Living Conditions in China’s Megacities” in 2019. This institute proposed preliminary evaluation indicators through a literature review, and revised and weighted these indicators through expert consultation to form a questionnaire on individual living conditions, subjective quality, and mental health in mega-cities. This institute conducted the survey in ten Chinese mega-cities, namely Beijing, Tianjin, Shanghai, Hangzhou, Guangzhou, Shenzhen, Chongqing, Chengdu, Wuhan, and Changsha. This study also collected information on the housing conditions and community environments of the respondents using the website Anjuke (www.anjuke.com, accessed on 20 April 2020). This study used the survey data and online data to analyze the correlations among housing conditions, subjective housing quality, and mental health, using multi-level regression models.

### 3.2. Setting and Participants

The Social Survey Center conducted the survey in ten Chinese mega-cities. In each mega-city, this institute randomly selected 40 urban communities, using stratified multi-stage cluster sampling; from each community, it also randomly selected about 13 adults aged from 18 to 65 years old to answer the questions on their housing conditions, subjective housing quality, and mental health. Research team members from the center conducted the face-to-face interviews with the participants in these communities. Consent was obtained from the participants. Participation was voluntary, anonymous, and confidential. Thus, this study obtained 5032 valid cases from a total of 406 communities in these 10 mega-cities for statistical analysis.

This study scraped open information on housing conditions and community environments from the website Anjuke (www.anjuke.com). This Chinese website is a well-known real estate information service platform that provides not only housing transaction services but also information on housing and communities, such as the average house prices in a neighborhood, the date of the neighborhood’s construction, plot ratios, etc. This study also collected the numbers of restaurants, kindergartens, elementary schools, junior high schools, hospitals, parks, and subway stations within a 1 km radius of each neighborhood using the Gaode map API interface, and calculated the shortest distance, respectively, from the neighborhoods to their respective city centers. The data used in this study and relevant documents can be downloaded from the website www.Figshare.com using the following DOI: https://doi.org/10.6084/m9.figshare.28142324 (accessed on 29 March 2025).

### 3.3. Variables

#### 3.3.1. Dependent Variable

The dependent variable was mental health, measured with a ten-item version of the Center for Epidemiologic Studies’ Depression Scale (CES-D-10). It is a widely used screening scale for depression ([22]). Scholars have previously used this checklist to screen for mental distress in a Chinese context ([29]). Scores range from 0 to 30, with a higher score indicating greater mental health. The Cronbach’s alpha is 0.79 for this sample.

#### 3.3.2. Mediating Variable

Subjective housing quality is conceptualized as a continuous variable. The questionnaire explores this variable using the following question: “In contemporary society, houses are often categorized as good or bad, luxurious or ordinary. If we use a scale of 1–10 to represent the housing stratification, where 1 signifies the poorest housing conditions and 10 represents the most desirable, where would you place the housing you are currently living in?”. The mediating variable in this study is based on this question; the higher the score assigned to this variable is, the higher the level of subjective housing quality will be.

#### 3.3.3. Independent Variables

The independent variables consisted of two categories: the individual-level housing status and the community-level neighborhood environment. Two indicators at the individual level were selected to measure housing status: home ownership and housing area.

For home ownership, a value of 0 denoted not owning a house or renting a flat, whereas a value of 1 denoted being a homeowner.

This study measured housing area by asking the following question: “What is the floor area (M^2^) of the home you are currently living in?”.

At the community level, the community environment included the neighborhood location, housing price, the age of the housing, plot ratio, the type of neighborhood, the neighborhood amenities, and the surrounding educational resources.

The neighborhood location was measured as the shortest distance (km) from the community to its respective city center.

Relative housing prices were derived from the ratio of the average house price in the community in question to the average housing price within the city as a whole.

The age of the housing was defined as the year in which the neighborhood was built.

The community type was categorized as either a commercial community (denoted as 1) or a non-commercial community (denoted as 0).

The plot ratio was defined as the ratio of the size of a plot of land to the amount of floor area that can be built on it, ranging from 0.4 to 9.46.

Other control variables also included age, sex, marital status, years of education, type of household, party membership, the number of other houses owned, the logarithm of annual household income, and the logarithm of household loans.

Table 1 defines the key variables and shows the results of the descriptive statistical analysis. In our sample, the average subjective housing quality in mega-cities is 5.927, indicating a moderately high self-assessed housing rating. The average age of the sample is approximately 44.9 years. The gender ratio is almost balanced, with 46% being female. The average number of years of education is more than 12, indicating that most residents of mega-cities have finished high school. More than 76% of the respondents are married. A total of 73.4% of them have a local household registration, and approximately 17% of them are members of a political party. The average house area exceeds 90 square meters, and the proportion of those with home ownership is 69.4%, indicating that 30.6% of residents rent or borrow housing. The average number of other houses owned is 0.38, demonstrating that the majority of residents in mega-cities own one house and do not own another. The average age of housing is nearly 20 years. More than 60% of residents live in commercial communities. The average price per square meter of housing in these mega-cities is CNY 32,937.63, indicating that housing prices in mega-cities are generally high. The average distance between a residential community and its respective city center is nearly 18 km, indicating the importance of the spatial structure of these mega-cities.

### 3.4. Analysis Methods

This study tested the multi-level models proposed in the preceding paragraphs with survey data from 10 Chinese mega-cities, using Stata 17.0 for the statistical analyses. This study applied multi-level linear regression models to examine the cross-level influence of housing conditions on subjective housing quality and the moderating effect of home ownership on the relationship between housing conditions and subjective housing quality, due to the two-level structure of the data ([15]). All variables were centered.

A null model was then constructed to estimate the variance distribution of subjective housing quality at the individual level and community level. Intra-class correlation coefficients (ICCs) were used to assess the extent to which neighborhood-level variance is explained as a proportion of total variance. Subsequently, this study used stepwise multi-level regression models to analyze the extent to which the cross-level effects of housing environments, respectively, affect subjective housing quality and mental health among the residents. Multi-level regression models have previously been used to examine the structural effects of community factors on individual consequences ([3]; [9]). Next, this study included an interaction term between home ownership and subjective housing quality to test whether differences in home ownership change the relationship between subjective housing quality and mental health.

Finally, this study applied the Sobel–Goodman test ([26]) to examine the mediating effects of subjective housing quality on the relationships between housing conditions and mental health. First, we regressed the dependent variable, mental health, on housing conditions, along with the control variables. The coefficients for the housing conditions variable were the total effects. Second, we regressed the mediating variable, subjective housing quality, on the housing conditions variable and the control variables. Third, we regressed the dependent variable, mental health, on the housing conditions, subjective housing quality, and the control variables. This study used the Sobel–Goodman test to calculate the coefficients of the aforementioned mediating effects ([19]).

## 4. Results

In Table 2, the results of the null model, including only the subjective housing quality variable, show that the between-group correlation (ICC) is 0.187, indicating that as much as 18.7% of the variance in subjective housing quality can be explained at the community level. The community environments of these neighborhoods have important effects on individual evaluations of their housing status. Thus, we need to take community characteristics into account.

Model 1 incorporates housing area and home ownership into the multi-level linear regression model, controlling for age, gender, year of education, marital status, household registration, political party membership, number of other houses owned, household income, household loan, and the dummy variables of the respective cities. The results of this model show that the ICC declines significantly from 0.19 to 0.087. Home ownership is highly correlated with subjective housing quality. Compared with non-homeowners, urban residents living in their own houses have a 0.415-unit advantage in terms of subjective housing quality (*p* < 0.001).

Housing area is strongly correlated with subjective housing quality. On average, an additional square meter of living space is associated with a 0.008-unit increase in subjective housing quality; the results also indicate that this association is significant. A larger house is more comfortable, as more rooms mean that the family members’ requirements of privacy and comfort can be fulfilled. These two variables are significantly related to subjective housing quality at the individual level.

In addition, the unexplained variance at the community level decreases from 0.722 in the null model to 0.333 in Model 1, and the unexplained variance at the individual level decreases from 3.083 to 2.947. ICC decreases from 0.19 to 0.087, indicating that the inclusion of the housing conditions at the individual level substantially reduces the unexplained variance.

Model 2 includes community-level variables in the multi-level regression model. Community educational resources, surprisingly, have negative effects on subjective housing quality (β = −0.114, *p* < 0.05). This significant cross-level effect implies that more educational resources surrounding a neighborhood may reduce individual subjective housing quality.

Respondents in newer communities give more positive evaluations to their housing status. A one-year increase in the age of neighborhood housing decreases the level of subjective housing quality by 0.015 units (*p* < 0.001), suggesting that living in a new community can help improve the subjective housing quality. The building materials used in old houses are limited, and the quality of these houses declines as their age increases. There are many irrationalities in the room designs of older neighborhoods; additionally, the surrounding amenities are prone to deterioration, and services such as property management, vehicle parking, etc., are much poorer. Thus, the age of neighborhood housing significantly reduces the subjective housing quality of residents ([28]).

A community with a higher plot ratio tends to have residents who perceive their housing quality as being higher. (β = 0.087, *p* < 0.01). This implies that the density of communities is highly related to housing quality. Residents in Chinese mega-cities tend to live in more crowded neighborhoods due to the preference for urban-collective ways of living and therefore have more chances to communicate or interact with their neighbors ([28]). These facts may increase their subjective evaluations of housing in their neighborhoods.

In addition, there are no significant correlations of subjective housing quality with the distance to the city center, relative house price, community type, and community amenities.

The age of the housing indicates the conditions of the physical materials used within the neighborhood, in terms of architecture and the neighborhood environment. The better these conditions are, the stronger the communities’ residential functions are, and the better the communities can meet residents’ needs. However, educational resources are negatively correlated with subjective housing quality. Further exploration of the specific mechanisms of these negative effects is needed.

Table 3 reports the fitting results from multi-level models between mental health and subjective housing quality ([15]). Model 3 includes all the independent variables and control variables. Both housing area and home ownership are positively associated with mental health. Only the distance to the city center is negatively related to mental health, while all other community variables have no significant effects on mental health.

Model 4 indicates that the regression coefficient for subjective housing quality is statistically significant (β = 0.264, *p* < 0.001), while housing area has no significant effect on mental health after controlling for subjective housing quality. These results suggest that subjective housing quality could fully mediate the effect of housing area on mental health ([2]).

In Model 5, the interaction term between home ownership and subjective housing quality is significantly positive (β = 0.214, *p* < 0.01), suggesting that subjective housing quality can increase homeowners’ mental health more than renters’ mental health.

Model 6 examines the effect of subjective housing quality on mental health within the subgroup of homeowners. The coefficient of subjective housing quality (β = 0.343, *p* < 0.001) is much greater than that in Model 4.

Finally, Model 7 tests the multi-level model with the subgroup of renters. The coefficient of subjective housing quality is only 0.059, suggesting that subjective housing quality does not affect renters’ mental health. In other words, subjective housing quality does not serve as an intermediary factor between housing conditions and mental health among renters.

According to [2]’s ([2]) argument, the results from multi-level models satisfy the premise of the mediating effect analysis. This study applies Sobel–Goodman tests ([26]) to determine the significance of the mediating effects, the results of which are reported in Table 4. The effect of home ownership on mental health is partly mediated by subjective housing quality. The indirect effect of home ownership makes up 18.7% of the total effect. The indirect effect of housing area is largely mediated by subjective housing quality, with 50% of the total effect being mediated. Plot ratio, the age of the housing, and educational resources have indirect effects on mental health, but their total effects are statistically insignificant. Location only has a direct effect on mental health.

## 5. Discussion

The rapid growth in real estate development exacerbates housing inequality in China’s mega-cities. This study tries to reveal the relationship between housing conditions and residents’ evaluations of their housing status and their mental health. Using data from ten Chinese mega-cities, this empirical study indeed shows that the relationship between housing conditions and mental health is significantly mediated by subjective housing quality.

First, both of the key indicators of housing conditions—home ownership and housing area—are positively related to mental health, which is consistent with previous studies conducted in China ([4]; [30]). Subjective housing quality plays an important mediating role in the relationship between housing conditions and mental health. Undoubtedly, home ownership is an important factor in determining subjective housing quality and mental health in China’s mega-cities, and this result is also consistent with the findings of previous studies ([1]; [7]). Owning a house provides residents with a sense of home and acts as a buffer against negative events and economic risks. Homeowners are also willing to invest more in their own homes and in consistently improving the quality of their living conditions ([1]; [12]).

Moreover, this study also confirms the importance of housing area for subjective housing quality and mental health. It is well known that larger houses provide people with more comforts, privacy, and convenience. When assessing their housing conditions, residents prioritize not just home ownership, but also comfort and convenience ([20]). Renting a larger flat may provide renters with higher levels of subjective housing quality.

Second, neighborhood environments are significantly related to subjective housing quality. Recently constructed communities often feature more scientifically and properly designed houses, built with more durable materials, that have been developed, built, and sold by real estate developers. Such communities boast superior construction quality. Furthermore, these communities also offer better property management, thereby enhancing residents’ living experiences ([29]). Older neighborhoods often suffer from inadequate housing quality and increased maintenance issues, and, in certain instances, rely on public housing authorities for their management. As a result, residents living in these new communities have higher levels of subjective housing quality than those in older communities.

Plot ratio is also positively associated with subjective housing quality but not with mental health. In Hong Kong, living in communities with high plot ratios may cause psychological distress, but this negative effect only exists in apartments shared by multiple family units ([21]). In China’s mega-cities, a higher plot ratio means a higher population density and better community services ([33]). Thus, plot ratio is not directly associated with residents’ mental health, but its association with subjective housing quality is still notable.

However, to some extent, educational resources are negatively related to subjective housing quality. In mega-cities, the concentration of educational resources may raise residents’ subjective evaluations of housing quality and reduce their evaluations of their housing status ([16]). Recently, the pace of urban construction and expansion in mega-cities has not kept up with the redistribution of schools and teachers. Most good educational resources are still concentrated in the downtown or urban core areas. These areas are characterized by dense population, severe traffic congestion, and subpar living experiences ([17]; [33]). Many residents are fed up with these problems and thus have lower levels of subjective housing quality.

In addition, location is directly associated with mental health. Downtown neighborhoods usually provide excellent transportation links, lifestyle amenities, and vibrant commercial scenes. The residents in these housing enjoy greater convenience for shopping and commuting. In China’s mega-cities, most high-quality public service resources are concentrated in the city center or nearby; therefore, suburban communities often grapple with challenges like inadequate transportation, lengthy commutes, and difficulties in shopping. The disadvantages of suburban housing in terms of convenience may not be immediately apparent in residents’ personal assessments of their housing, but they can profoundly undermine residents’ mental health.

Third, home ownership significantly moderates the association between subjective housing quality and mental health; specifically, home ownership strengthens the association of housing with mental health. When individuals reside in their own homes, this implies greater wealth accumulation and enhanced security. Renters are often faced with poor housing quality and high mobility ([4]). Compared to homeowners, renters are less mentally attached to the housing they live in ([12]; [30]). Consequently, the mediating effect of subjective housing quality is insignificant among renters.

Recently, China’s mega-cities have scaled down their large-scale demolition and construction endeavors, opting instead for numerous urban renewal initiatives aimed at enhancing residents’ quality of life ([32]). Therefore, in the coming years, older communities and disadvantaged communities in the mega-cities will witness significant improvements in living standards, thereby narrowing the housing disparities among various communities within mega-cities, ultimately improving subjective housing quality and mental health.

There are a few limitations in this study. Firstly, in terms of sample selection, this study is based on data from ten well-developed mega-cities in China. Thus, these findings may not be generalizable to the whole country, especially to the small and medium-sized cities. Secondly, this study uses data from a cross-sectional survey, precluding the assessment of causal links between housing conditions, subjective quality, and mental health. Further studies should address endogenous problems between housing conditions, subjective housing quality, and mental health. Some control variables in this study also have a strong correlation with mental health and subjective housing quality. Further in-depth research is therefore needed to explore the causal relationships between them ([18]). Thirdly, the questionnaire provides limited information on the objective quality of housing and on community services. There is a notable lack of investigation into community characteristics such as community culture, community social interactions, and community governance. Further studies are needed to explore issues such as the cross-level mechanism through which community social capital and community governance impact residents’ subjective housing quality and mental health.

## 6. Conclusions

This study demonstrates that, for a sample of adults from ten Chinese mega-cities, housing conditions are important factors for understanding the variations in subjective housing quality and mental health. Both housing area and home ownership are positively related to subjective housing quality and mental health in these mega-cities. Regarding the neighborhood environment, educational resources and age of housing are negatively and saliently associated with subjective housing quality, while plot ratio is significantly positively related to it. Location as a variable is directly related to residents’ mental health. Importantly, the association between subjective housing quality and mental health is much stronger among homeowners than among renters.

## Figures and Tables

**Table 1 behavsci-15-00485-t001:** Descriptive statistics for key variables (N = 5032).

Variable	Description	Average Value	Standard Deviation
Mental health	Continuous variable, 1–30	23.949	4.469
Subjective housing quality	Continuous variable, 1–10	5.923	1.959
Age	Continuous variable, 18–65 (years)	44.918	13.74
Gender	Dummy variable, male = 1, female = 0	0.462	0.499
Years of education	Continuous variable, 0–19 (years)	12.541	3.824
Marital status	Dummy variable, married = 1, unmarried = 0	0.766	0.425
Household registration	Dummy variable, municipal household registration = 1, non-local household registration = 0	0.737	0.442
Political party membership	Dummy variable, Communist Party of China (CPC) member = 1, non-CPC member = 0	0.168	0.374
Household income	Continuous variable, logarithm of annual household income	11.432	2.065
Household loans	Continuous variable, logarithm of household loans	2.077	4.518
Housing area	Continuous variable, the area in which respondents currently live	90.59	48.487
Home ownership	Dummy variable, home ownership = 1, no home ownership = 0	0.695	0.461
Number of other houses	Continuous variable, number of other dwellings currently owned by the household	0.378	0.906
Age of housing	Continuous variable, the earliest age of the housing in the neighborhood	19.858	13.523
Community type	Dummy variable, commercial community = 1, non-commercial community = 0	0.555	0.497
Relative housing price	Continuous variable, average price per square meter of housing in the community	32,937	27,795
Community amenities	Continuous variable, principal component factor extracted from the logarithm of the number of restaurants, the number of hospitals, parks, gyms, and subway stations	0.002	1.005
Educational resources	Factor extracted from the number of kindergartens, elementary schools, middle schools, high schools, and universities surrounding the neighborhoods	−0.006	0.995
Plot ratio	Continuous variable, the ratio between the gross floor area of a building and the area of the site on which it is erected	2.23	1.175
Community location	Continuous variable, the shortest distance between the community and the city center	18.872	26.785

**Table 2 behavsci-15-00485-t002:** Results of multi-level linear regressions on the effects of housing and community environments on subjective housing quality (N = 5032).

Variable	Null Model	Model 1	Model 2
Housing area		0.008 ***	0.008 ***
		(0.059)	(0.001)
Home ownership		0.415 ***	0.408 ***
		(0.064)	(0.064)
Relative house price			−0.003
			(0.002)
Community type			0.069
			(0.088)
Age of housing			−0.014 **
			(0.005)
Plot ratio			0.079 **
			(0.035)
Community amenities			0.002
			(0.046)
Educational resources			−0.109 *
			(0.047)
Community location			−0.124
			(0.248)
Random-effects parameters			
Between-group variance	0.722	0.282	0.245
	(0.06)	(0.038)	(0.035)
Within-group variance	3.083	2.972	2.97
	(0.045)	(0.062)	(0.062)
ICC	0.19	0.087	0.076

Note: These models also control for age, gender, year of education, marital status, household registration, political party membership, number of other houses, household income, household loan, and the dummy variables of the respective cities. Levels of statistical significance: * *p* < 0.05, ** *p* < 0.01, *** *p* < 0.001.

**Table 3 behavsci-15-00485-t003:** Results of multi-level linear regressions on the effects of subjective housing quality on mental health.

	Model 3	Model 4	Model 5	Model 6	Model 7
Subjective housing quality		0.264 ***	0.113	0.343 ***	0.059
		(0.034)	(0.059)	(0.040)	(0.065)
Housing area	0.003 *	0.001	0.001	0.001	0.004
	(0.001)	(0.001)	(0.001)	(0.002)	(0.003)
Home ownership	0.568 ***	0.461 **	0.533 ***	-	-
	(0.155)	(0.155)	(0.156)	-	-
Relative house price	−0.002	−0.002	−0.002	−0.008	0.002
	(0.005)	(0.005)	(0.005)	(0.006)	(0.008)
Community type	−0.026	−0.045	−0.041	0.043	−0.225
	(0.214)	(0.215)	(0.214)	(0.224)	(0.369)
Age of housing	−0.008	−0.004	−0.004	−0.006	−0.008
	(0.011)	(0.011)	(0.011)	(0.012)	(0.017)
Plot ratio	−0.070	−0.091	−0.090	−0.052	−0.153
	(0.085)	(0.085)	(0.085)	(0.089)	(0.143)
Community amenities	0.057	0.086	0.085	0.183	−0.058
	(0.112)	(0.112)	(0.112)	(0.118)	(0.197)
Educational resources	−0.133	−0.133	−0.139	−0.120	−0.156
	(0.111)	(0.111)	(0.111)	(0.118)	(0.184)
Location	−1.576 **	−1.544 *	−1.581 **	−1.450 *	−3.086 *
	(0.603)	(0.605)	(0.602)	(0.605)	(1.244)
Home ownership * Subjective housing quality			0.214 **		
			(−0.068)		
Random-effects parameters					
Between-group variance	1.537	1.479	1.458	1.067	2.491
	(0.209)	(0.205)	(0.203)	(0.219)	(0.523)
Within-group variance	17.267	17.265	17.241	16.864	17.513
	(0.359)	(0.359)	(0.358)	(0.427)	(0.709)
N	5032	5032	5032	3498	1534
ICC	0.082	0.079	0.078	0.059	0.125

Note: These models also control for age, gender, year of education, marital status, household registration, political party membership, number of other houses, household income, household loan, and the dummy variables of the respective cities. Levels of statistical significance: * *p* < 0.05, ** *p* < 0.01, *** *p* < 0.001.

**Table 4 behavsci-15-00485-t004:** Results of Sobel–Goodman tests on the mediating effects of subjective housing quality on mental health (N = 5032).

Independent Variable	Indirect Effect		Direct Effect		Total Effect	
	Coef.	S.E.	Coef.	S.E.	Coef.	S.E.
Home ownership	0.103 ***	−0.023	0.449 **	−0.163	0.552 ***	−0.163
Housing area	0.002 ***	0	0.002	−0.002	0.004 *	−0.002
House price	−0.001	0	−0.002	−0.004	−0.002	−0.004
Plot ratio	0.020 **	−0.007	−0.088	−0.064	−0.068	−0.063
Age of housing	−0.004 ***	−0.001	−0.002	−0.008	−0.006	−0.008
Educational resources	−0.025 *	−0.011	0.068	−0.084	0.043	−0.085
Community type	0.02	−0.017	−0.032	−0.159	−0.012	−0.159
Community amenities	−0.002	−0.008	−0.111	−0.079	−0.113	−0.079
Location	−0.045	−0.05	−1.525 ***	−0.44	−1.570 ***	−0.448

Note: Levels of statistical significance: * *p* < 0.1, ** *p* < 0.05, *** *p* < 0.01.

## Data Availability

Data collected and analyzed during this study can be downloaded from the website of www.Figshare.com with the DOI (https://doi.org/10.6084/m9.figshare.28142324, accessed on 29 March 2025).

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
