# Peer review of "The Mediating Effect of Subjective Housing Quality on the Relationship Between Housing Conditions and Mental Health: Evidence from China’s Mega-Cities"

_behavsci, 2025, doi:10.3390/bs15040485_

Round 1
Reviewer 1 Report
Comments and Suggestions for Authors
Thank you for inviting me to review this work. I found it clear and easy to follow. The issue of housing and what contributes to 'good housing' is an important area of study as housing availability, affordability and adequacy are issues in many countries. This study is unique in that it addresses the issue of housing and its relation to mental health, an important area for research.
As for comments, there are several places in the text where I think information should be added:
- Line 259/60 - supporting reference
- Line 278/79- supporting reference
- Line 291/92 - what impact does this have? How useful is it to know that for every one unit of housing quality mental health increases by 0.343 units?
- Line 335-337- supporting reference
- Line 360-61 How do you know this? More information is needed
- Line 375- supporting reference
- Line 381 Perhaps 'may be' rather than "likely". Likely infers more certainty, so if that is true, you need more support for it here
I find the conclusion very strong. I think the addition of supporting references suggested above could add further strength to a good paper.
Thanks again for asking me to review this paper and good luck in your future work!
Author Response
- Comment 1: Line 259/60 - supporting reference
Response 1: I add a citation in Line 280 and Line286: Wang, H. & Ruan, X. 2022. The Misconceptions of High-Rise Housing: Rethinking China’s Housing Patterns. Architectural Journal, 6: 82-89.
- Comment 2: Line 278/79- supporting reference:
Response 2: I add a citation in Line 298: Hox, J., & Wijngaards-de Meij, L. (2014). The multilevel regression model. The SAGE Handbook of Regression Analysis and Causal Inference, 133–152. https://doi.org/10.4135/9781446288146.n7
- Comment 3: Line 291/92 - what impact does this have? How useful is it to know that for every one unit of housing quality mental health increases by 0.343 units.
Response 3: I delete the confusing statement “It means that in the subgroup of home purchasers, one unit of subjective housing quality can increase 0.343 unit of mental health.”
- Comment 4: Line 335-337- supporting reference
Response 4: I add two citations: Baker, E., Bentley, R., & Mason, K. (2013). The mental health effects of housing tenure. Urban Studies, 50(2), 426-442. ; Grimes, A., Smith, C., O’Sullivan, K., Howden-Chapman, P., Le Gros, L., & Dohig, R. K. (2024). Housing Tenure and Subjective Wellbeing: The Importance of Public Housing. Applied Research in Quality of Life, 1-24.
- Comment 5: Line 360-61 How do you know this? More information is needed
Response 5: Response: I add two citations in Line 376 and Line 286: You, H., & Yang, X. (2017). Urban expansion in 30 megacities of China: Categorizing the driving force profiles to inform the urbanization policy. Land Use Policy, 68, 531-551.; Huang, L., Yan, L., & Wu, J. (2016). Assessing urban sustainability of Chinese megacities: 35 years after the economic reform and open-door policy. Landscape and Urban Planning, 145, 57-70.
- Comment 6: Line 375- supporting reference
Response 6: I add two citations in Line 403: Grimes, A., Smith, C., O’Sullivan, K., Howden-Chapman, P., Le Gros, L., & Dohig, R. K. (2024). Housing Tenure and Subjective Wellbeing: The Importance of Public Housing. Applied Research in Quality of Life, 1-24; Wang, S., Cheng, C., & Tan, S. (2019). Housing determinants of health in urban China: A structural equation modeling analysis. Social Indicators Research, 143, 1245-1270.
- Comment 7: Line 381 Perhaps 'may be' rather than "likely". Likely infers more certainty, so if that is true, you need more support for it here
Response 7: I revise the sentence according to the comment from Line 407 to Line 411 with the statement “older communities and inconvenient communities in the mega-cities will witness the significant improvement in living standards, thereby narrowing the housing disparities among various communities within mega-cities, ultimately improving subjective housing quality and mental health”.
Reviewer 2 Report
Comments and Suggestions for Authors
Please see attached file.

There are a few typos and several unclear sentences, including the usage of "housing tenure" as a synonym for "home-ownership".
Author Response
Comment 1 The main issue I see with the manuscript as it stands is that the survey used, that I understand is collected directly by the author, is not described in detail in terms of procedures, timings, sample selection, questions. I suggest that a separate Appendix is used for this. Moreover, to favour transparency and replicability, I would encourage the data and replication code to be made available. Since these are new data, making all these resources available will also increase usage of the data and citations for the author.
Response 1: I upload the data and relevant documents about the sampling and the questionnaire to an open platform. The readers can find the data and code with the DOI given in the article (Line 146).
Comment 2 : Another major issue is the usage of causal terminology when the analysis is not credibly causal, because it does not isolate exogenous variation in the independent variables, nor eliminate, through the regression model, all omitted variable bias. Although the author says in the limitations that the analysis is not meant to be causal, everywhere else in the manuscript the language is causal and should be adjusted. For instance, from the abstract: “housing area and housing tenure exert significant positive effects on subjective housing quality”. As a follow-up point, I am unconvinced that there is enough evidence in the manuscript to say that subjective housing quality mediates the effect of housing conditions on mental health. Leaving aside the empirical concerns regarding causality discussed above, it is unclear why it isn’t possible for instance that housing conditions affect both subjective housing quality and mental health directly, without the mediating effect of one on the other.
Response2 : I revise the statements on casual effects of individual variables, like housing area and housing ownership, on mental health, according to the comment. (In Line 18, 43, 51, 124, 223, 258, 348, 397, 419). In addition, I also add two citations to clarify the cross-level effect in the multilevel models: Blakely, T. A., & Woodward, A. J. (2000). Ecological effects in multi-level studies. Journal of Epidemiology & Community Health, 54(5), 367-374. (Line 220); DiPrete, T. A., & Forristal, J. D. (1994). Multilevel models: methods and substance. Annual review of sociology, 20(1), 331-357.(Line 221)
Comment 3 :I list below a series of additional considerations: - In the abstract the sentence “The results from multilevel linear regression models show that housing areas and housing tenure exert significant positive effects on subjective housing quality” is ambiguous concerning the direction of the effect. It says positive, but housing areas and housing tenure per se don’t have a direction.
Response 3: I revise the part of the abstract, according to the comment. (In Line 11-12)
Comment 4: It would be good to have a reference whenever a precise claim is made, such as at lines 26 and 27 of the manuscript or lines 113-115
Response 4: I add two citations according to the comments. Huang, L., Yan, L., & Wu, J. (2016). Assessing urban sustainability of Chinese megacities: 35 years after the economic reform and open-door policy. Landscape and Urban Planning, 145, 57-70. ; Hu, M., Lin, Z., & Liu, Y. (2024). Housing disparity between homeowners and renters: Evidence from China. The Journal of Real Estate Finance and Economics, 68(1), 28-51.
Comment 5: Lines 68-69: very old reference if one wants to claim that very few analyses on the topic exist. (Henretta, 1984)
Response 5 : I include two citations in Line 63, according to the comment.
Comment 6: Lines 72-77: the meaning is unclear. Housing tenure seems to be used as meaning “Home-ownership”, but home-ownership is just one possible housing tenure (see also lines 157-158).
Response 6: I replace tenure with ownership accordingly in Line 17, 21, 34, 82, 108, 167, 195, 210, 223, 242, 246, 248, 300, 315, 330, 331, 345, 349, 359, 397, 398, 431.
Comment 7: Again, it is said that “housing conditions are positively associated”, but housing conditions per se don’t have a direction.
Response 7 : I revise this sentence accordingly. (In Line80-81)
Comment 8: - When discussing urban communities, it seems that they have specific administrative features. It would be useful to have more background on this, as well as on the possible types of commercial and non-commercial (lines 169-170).
Response 8: unfortunately, this survey does not have any other information on community administrative features. The real estate website has no details either.
Comment 9: - Line 150: I think the author mean mediator variable and not dependent?(mediating variable)
Response 9: Yes, it is a big mistake. I correct it accordingly. Many thanks for the reviewer. (In Line 161)
Comment 10- Table 1 and line 199 (other instances later in the manuscript): N is equal to 5032, but in Section 3.1 earlier, it says that there are 9603 valid cases.
Response 10 : Yes, these are mistakes too. I correct them accordingly. Many thanks for the reviewer. (In Table 1 and Line 134)
Comment 11- Line 220: It would be useful to specify which control variables the author is referring to (presumably the person-level variables described earlier). This also applies to the note of Table 2. (List control variables)
Response 11: I specify the control variables in Line 244, 245 and 246 and also add the list of control variables to the notes of Table 2 and Table 3.
Comment 12- Lines 245-246: On top of the general comment about causal claims, I would be especially cautious about causal claims involving control variables. See: 2 Hünermund, P., & Louw, B. (2025). On the nuisance of control variables in causal regression analysis. Organizational Research Methods, 28(1), 138-151.
Response 12: I revise the sentences from Line 243 to 250 and add the limitation of this study in Line 420.
Comment 13: - Models 6 and 7 in Table 3: The N of these subsets of the sample should be reported.
Response 13: I add the N and the numbers of sub-samples in Table 3.
Comment 14 :- Lines 300-301: another example of unclear sentence. (deleted)
Response 14 : I delete this unclear sentence according to the comment.
Comment 15 :- Line 303: It would be useful to briefly describe the mechanics of the Sobel test
Response 15 : I add a paragraph briefly describe the Sobel test of the mediating effect from Line 226 to 234 and introduce a citation: MacKinnon, D. P., Lockwood, C. M., Hoffman, J. M., West, S. G., Sheets, V.(2002). A comparison of methods to test mediation and other intervening variable effects. Psychol Methods, 7(1):83-104.
Comment 16 :- Line 336: Unclear what “improving the matching degree” mean (revised )
Response 16: I revise the sentence according to the comment. (Line 360, 361)
Round 2
Reviewer 2 Report
Comments and Suggestions for Authors
The author has put effort in addressing the concerns I had expressed and this is appreciated. I still have two main remarks:
- I could not find the data on FigShare with the DOI provided. Are the data not available yet? It may also be my mistake, not being familiar with the platform, but it should be ensured that they are actually accessible.
- There are still many instances of use of causal language, in the sense that it is claimed that a variable has an effect on another. This implies a link of causality.
The quality has improved but there are still instances of repetitions and sentences that are not clearly structured. The writing would benefit from further review and refinement.
Author Response
Comments 1: The author has put effort in addressing the concerns I had expressed and this is appreciated. I still have two main remarks: I could not find the data on FigShare with the DOI provided. Are the data not available yet? It may also be my mistake, not being familiar with the platform, but it should be ensured that they are actually accessible.
|
Response 1: I have double checked the files on the website of www.Figshare.com. It has been viewed and downloaded 38 times. I uploaded the screenshot of the webpage of the Figshare.com |
Comments 2: There are still many instances of use of causal language, in the sense that it is claimed that a variable has an effect on another. This implies a link of causality. |
Response 2: Agree. I have, accordingly, revised the manuscript to emphasize the correlations among the variables (in Line 15, 39, 368, 436 and 438). The revisions are in red. |
Response to Comments on the Quality of English Language |
Point 1: The quality has improved but there are still instances of repetitions and sentences that are not clearly structured. The writing would benefit from further review and refinement. |
Response 1: I used the MDPI English editing service to improve the Quality of English Language. |